# Moisture Content Impact on Properties of Briquette Produced from Rice Husk Waste

**Anwar Ameen Hezam Saeed [1]**, **Noorfidza Yub Harun [1,*]**, **Muhammad Roil Bilad [2]**, **Muhammad T. Afzal [3]**, **Ashak Mahmud Parvez [3,4]**, **Farah Amelia Shahirah Roslan [1]**, **Syahirah Abdul Rahim [1]**, **Vimmal Desiga Vinayagam [1]** and **Haruna Kolawole Afolabi [1]**

[1] Department of Chemical Engineering, Universiti Teknologi PETRONAS, Seri Iskandar 31750, Perak, Malaysia; anwar_17006829@utp.edu.my (A.A.H.S.); farah_20000280@utp.edu.my (F.A.S.R.); syahirah_19000130@utp.edu.my (S.A.R.); Vimmal_16002431@utp.edu.my (V.D.V.); haruna_18002856@utp.edu.my (H.K.A.)

[2] Department of Chemistry Education, Universitas Pendidikan Mandalika (UNDIKMA), Jl. Pemuda No. 59A, Mataram 83126, Indonesia; muhammadroilbilad@ikipmataram.ac.id

[3] Department of Mechanical Engineering, University of New Brunswick, 15 Dineen Drive, Fredericton, NB E3B 5A3, Canada; mafzal@unb.ca (M.T.A.); Ashak.Parvez@unb.ca (A.M.P.)

[4] Institute of Combustion and Power Plant Technology (IFK), University of Stuttgart, Pfaffenwaldring 23, D-70569 Stuttgart, Germany

* Correspondence: noorfidza.yub@utp.edu.my

**Abstract:** An agricultural waste-based source of energy in the form of briquettes from rice husk has emerged as an alternative energy source. However, rice husk-based briquette has a low bulk density and moisture content, resulting in low durability. This study investigated the effect of initial moisture contents of 12%, 14%, and 16% of rice husk-based briquettes blended with 10 wt% of kraft lignin on their chemical and physical characteristics. The briquetting was done using a hand push manual die compressor. The briquette properties were evaluated by performing chemical (ultimate and proximate analysis, thermogravimetric analysis), physical (density, durability, compressive strength, and surface morphology) analyses. The durability values of all briquette samples were above 95%, meeting the standard with good compressive strength, surface morphology, and acceptable density range. The briquette made from the blend with 14% moisture content showed the highest calorific value of 17.688 MJ kg$^{-1}$, thanks to its desirable morphology and good porosity range, which facilitates the transport of air for combustion. Overall, this study proved the approach of enhancing the quality of briquettes from rice husk by controlling the moisture content.

**Keywords:** biomass briquette; agricultural waste; energy; moisture content; rice husk; lignin; blend

## 1. Introduction

Supplying the demanded energy has been a serious problem for many developing countries due to set-off improvised technology and increasing population [1]. This promotes the utilization of multi-source fuels, particularly from sustainable and eco-friendly resources, including biomass. Research on biomass utilization as a fuel focuses on the utilization of biomass mainly form agro-waste for producing a solid fuel block called a briquette. It is considered as one of the available options to off-set the widening gap between the demand and the supply of energy, and considering the depleting nature of non-renewable energy sources from fossil fuels [2,3].

As the major Malaysian food crop, the annual production of rice husk, a side product of paddy mill processing, recorded by the Malaysia Ministry of Agriculture is 408,000 metric tons [4,5]. Rice husk is the outer layer of rice grain obtained from the paddy milling process [6]. The product obtained from rice consists of 72% grain, 5–8% bran, and 20–22% rice husk [7]. Generally, rice husk consists of 40% lignocellulose and cellulose, 5% hemicellulose as five-carbon sugar polymers, and the rest is minerals such as silica, alkali, and trace elements [8].

Table 1 shows a typical analysis of rice husk, as they vary according to climate condition, soil chemistry, and even the type of rice produced [9]. Rice husk is often dumped on landfills, which causes storage and operation problems. Additionally, poor management of rice husk causes serious environmental problems, as it is typically burned in open air [10,11].

**Table 1.** Typical analysis of rice husk [12].

| Property | Range | Basis |
|---|---|---|
| Bulk density (kg/m$^3$) | 96–160 | dry |
| Length of the husk (mm) | 2.0–5.0 | dry |
| Hardness (Mohr's scale) | 5.0–6.0 | dry |
| Ash (%) | 18–29 | dry |
| Carbon (%) | 35.0–42.0 | dry |
| Hydrogen (%) | 4.0–5.0 | dry |
| Oxygen (%) | 31.0–37.0 | dry |
| Nitrogen (%) | 0.23–0.32 | dry |
| Sulphur (%) | 0.04–0.08 | dry |
| Moisture content (%) | 6.0–10.0 | As received |

The use of rice husk as a base material for fuel briquette production has been shown to be effective in overcoming its negative impacts [13]. However, rice husk with low moisture content fails to produce a durable briquette. Thus, the high ash content becomes clogged in the boiler system during combustion, resulting in equipment failure and a high cost of maintenance. Moreover, agricultural residue-based briquette possesses a low quality, due to low lignin content [14,15]. The use of the bulk of biomass residue as fuel can minimize the wastage of valuable products by obeying the 3R concept (reduce, reuse, and recycle) [16].

The quality of a briquette greatly depends on its physical and chemical properties, which dictate its durability and its energy value [17]. Processing variables such as pressure, temperature, particle size, binding ratio, and moisture content massively affect the quality of the resulting briquettes [18,19]. This study explores the effect of moisture content on the resulting quality and durability of a briquette produced from rice husk. Briquettes were produced from rice husks with moisture contents of 12%, 14%, and 16%. Since rice husk was found to be low in moisture content and lignin, a kraft lignin binder was used to ease the compaction of the rice husk briquette [20,21]. After production, the rice husk briquettes were further characterized in terms of physical and chemical properties, with a focus on energy content (heating value) and standard durability. This study investigates the effect of initial moisture contents of 12%, 14%, and 16% of rice husk-based briquette blended with 10 wt% of kraft lignin on their chemical and physical characteristics, and demonstrates that rice husk has potential to be converted into highly durable briquettes.

## 2. Materials and Methods

### 2.1. Sample Preparation

The rice husk was obtained from the Bernas Kedah paddy mill, and the kraft lignin was obtained from Lembaga Getah, Malaysia. The raw rice husk was stored in a closed container under ambient temperature to prevent contamination. First, the rice husk was ground using an auto-grinder, followed by sieving with a mesh size of 500 μm. Each briquette was made from 10 g of blends, with a ratio of 90:10, and at different moisture contents of 12%, 14%, and 16%. The blending was done mechanically for 15 min. The moisture content of the blended biomass was measured using a Metler Toledo moisture content analyzer. The excess water content in the blend was removed by drying in an oven at a temperature of 105 °C, and the moisture drop was recorded regularly until the desired concentration was obtained. When the initial moisture content was lower than that aimed at, liquid water was sprayed in excess on the blend, and the process of adjusting moisture content via drying was performed until reaching the desired concentration. The blends

with desired moisture contents were then placed in a closed container to maintain their moisture content before processing.

### 2.2. Briquetting

The briquette samples were prepared in the form of a standard die at a diameter of 4 cm. The pressure was set constant at 150 MPa with 120 s holding time, and the briquetting was done at room temperature, 25 °C. The pressure level was selected based on a previous biomass briquetting study, in which the minimum pressure required was 150–200 MPa. The applied pressure, coupled with a proper moisture content feed, resulted in a briquette with desirable durability. Most researchers found that higher pressure tends to generate heat at elevated temperature from the die, which significantly melts the lignin content, whereby better compaction can be expected [22]. The compaction process was performed by using a hand-push manual pressure-based die compressor. The produced briquettes were stored in a closed container until further characterizations.

### 2.3. Physical Properties Characterization

#### 2.3.1. Density

The rice husk blend density was examined before and after briquetting to observe the density change. Three density parameters were tested which were tapped density, maximum density, and relaxed density. Tapped density was studied by inserting the rice husk blend in a cylindrical measuring tube and measuring the linear dimension (diameter and thickness) using a vernier caliper. The density was then calculated. The maximum density was calculated after the ejection of the briquette from the die compressor. Similarly, the relaxed density was measured after 7 days from the ejection day.

#### 2.3.2. Durability

A drop test was used to examine the briquette durability. The rice husk briquettes were dropped vertically from a height of 2 m, then the mass loss was weighed. The durability was calculated using Equation (1) [23]. This study is crucial, as it determines the quality and handling of briquette in terms of storage, transportation, and operation.

$$Durability = \frac{Mass\ of\ briquette\ before\ drop\ test}{Mass\ of\ briquette\ after\ the\ drop\ test} \tag{1}$$

#### 2.3.3. Compressive Strength

The compressive strength of rice husk briquette was tested according to the ASTM D575 method, using a universal testing machine [24]. A maximum load of 5 kN was applied diametrically on the briquette samples. This test was performed to observe the maximum load a briquette can withstand before any cracks or breaks appeared. The presence of the kraft lignin binder eased the compaction, and a detailed method on the preparation of the kraft lignin binder is available elsewhere [20].

#### 2.3.4. Surface Morphology

The surface morphology was obtained using scanning electron microscopy, mainly to observe the surface of the briquette in terms of a failure of the briquette after the maximum load was applied. Prior to measurement, the sample was dried at room temperature and coated with gold to avoid overcharging during the analysis at 15 kV. The contact between each particle can be observed through SEM testing, and the morphology can be used to justify the finest durability of the briquette sample.

### 2.4. Chemical Properties characterization

2.4.1. Ultimate Analysis and Proximate Analysis

The ultimate analysis of both the rice husk blend and the briquette was as determined by the element analyzer (CHNS analyzer ASTM D5291 Method) [24]. The proximate analysis of the rice husk blend and briquette was determined mainly on the higher heating value (HHV), ash content, and the presence of volatile matter, fixed carbon, and moisture content of the briquette. The higher heating value was determined by droperidol calorimeter. The ash content was determined by burning 2 g of briquette sample in a crucible placed in a furnace at a temperature of 700 °C for 3 h. The ash percentage (Ash %) was calculated by using Equation (2).

$$\text{Ash\%} = \frac{y(g) - x(g)}{m(g)} \times 100\% \qquad (2)$$

where y is the mass of the crucible lid with ash sample after analysis (g), x is the initial mass of the crucible lid (g), and m is the mass of the sample (g).

2.4.2. Thermogravimetric Analysis

The samples were heated until the maximum temperature of 800 °C with a heating rate of 20 °C/min. The degradation temperature of the briquette was identified through TGA and derivative thermogravimetric analysis (DTG) analysis. The degradation temperature is defined as the temperature with maximum weight loss. Subsequently, the fixed carbon, the volatile matter, and the moisture content of the rice husk blend and the briquette were determined from the TGA plot.

## 3. Results and Discussion

### 3.1. Briquette Compression

Figure 1 shows the rice husk briquette formed at various blend moisture contents, of 12%, 14%, and 16%. Since the particle size of the rice husk was less than 500 μm, the texture of the briquette obtained was smooth, which gives a better appearance [25]. The combinations of materials, blend ratio, applied pressure, and biomass size, as seen in Figure 1, gave the briquette samples a good form. A similar method of briquette compression was done by Ndindeng et al. [26], in which the briquetting was performed at a moisture content of 20%, usually after 7 days, for samples dried in the oven, and 21 days for samples dried under ambient conditions [27].

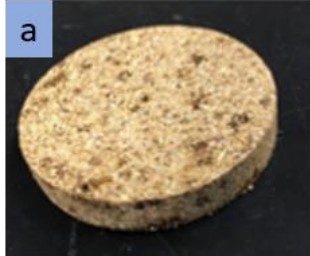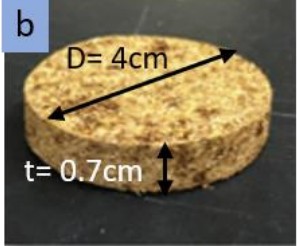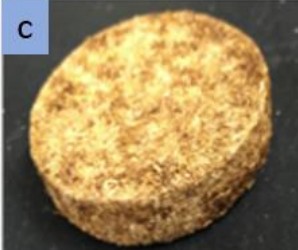

**Figure 1.** Pictures of the briquettes produced from rice husk at various moisture contents of (**a**) 12%, (**b**) 14%, and (**c**) 16%.

### 3.2. Physical Properties

3.2.1. Density

The properties of raw blends and briquettes at various moisture contents are summarized in Table 2. In general, the moisture content affects the density, the angle of friction, the force deformation characteristics, and the thermal conductivity, as reported elsewhere [28]. Based on the results obtained, the tapped density increased slightly at higher moisture contents. This can be attributed to the moisture contained in the blend. The maximum density obtained was calculated after the ejection of the briquette from the hand-push

manual compressor after 120 s of holding time. The briquette with maximum density was obtained with moisture content 16%, corresponding to a density of $1144 \pm 0.63$ kg/m$^3$, followed by 14% and 12%, with $1141 \pm 0.22$ kg/m$^3$ and $1136 \pm 0.67$ kg/m$^3$, respectively. The moisture content filled the micro gap in between the rice husk particles, connected by the kraft lignin binder. At high pressure, heat is generated at elevated temperatures in the die, which melts the binding agent and increases the compactness of the resulting briquette. Therefore, there is the possibility of fusing lignin more in the presence of higher moisture.

**Table 2.** Rice husk blend and briquette densities at various moisture contents.

| Moisture Content (%) | Tapped Density (kg/m$^3$) | Maximum Density (kg/m$^3$) | Relaxed Density (kg/m$^3$) |
|---|---|---|---|
| 12% | $144 \pm 0.66$ | $1136 \pm 0.67$ | $1136 \pm 0.67$ |
| 14% | $144 \pm 0.67$ | $1141 \pm 0.22$ | $825 \pm 0.26$ |
| 16% | $144 \pm 0.71$ | $1144 \pm 0.63$ | $673. \pm 0.17$ |

The relaxed density data show that the briquette made from the blend with the moisture content of 12% showed the highest density, followed by 14% and 16%, of $825 \pm 0.26$ kg/m$^3$ and $673.17 \pm 0.17$ kg/m$^3$, respectively. According to Mohsenin et al. [29], higher feedstock moisture content allows the briquette to expand, due to the higher residual stress after briquette ejection, causing elastic rebound or enlargement. A similar justification is applied in this study, as the density decreased with increasing moisture content. Moreover, a smaller particle size of biomass residue promotes a higher density under high pressures, as reported earlier [29]. The density of briquette increases with increasing moisture content until it reaches the maximum, and further increase in moisture content tends to reduce the density of briquette [25,30]. The standard briquette density falls within the range of 750 kg/m$^3$ and above so, based on these findings, briquette samples prepared from blends with moisture contents of 12% and 14% were identified as suitable, as they meet the standard briquette density requirement.High moisture content is able to make briquettes expand due to higher residual stress after ejection, which typically causes elastic rebound or enlargement [29,31].

3.2.2. Durability

Figure 2 shows durability data of the briquettes prepared from a blend with moisture contents of 12%, 14%, and 16% over 7 days of storage. They vary accordingly between intervals from day 0 until day 7. Durability percentage at moisture content 12% and 14% dropped only from 100% to 99%, and 100% to 98%, respectively, over 7 days, suggesting their long-term stability. However, the durability of the briquette from a blend with a moisture content of 16% dropped 1% in each day from day 0 to day 7, which can be attributed to the high moisture content in the briquette, which tends to reduce the durability, even though it enhances the burning efficiency and life span [32]. Typically, feedstock moisture content benefits the intermolecular and interfacial force which binds the particle by pressure and temperature generated by the die when compressed at high pressures [33]. The binding agent helps in fusing the biomass particles, and thus enhances the durability of the resulting briquette. Therefore, the durability of rice husk briquette at all moisture contents was above 96%, which falls within the standard pellet quality 95–100% durability range [34]. The results show that the standard durability of the biomass briquettes fell within the range of 95% and above. Moisture content is the main process parameter that significantly affects the durability. Most researchers found that lower durability often results in fines and hinders the operation and transportation.

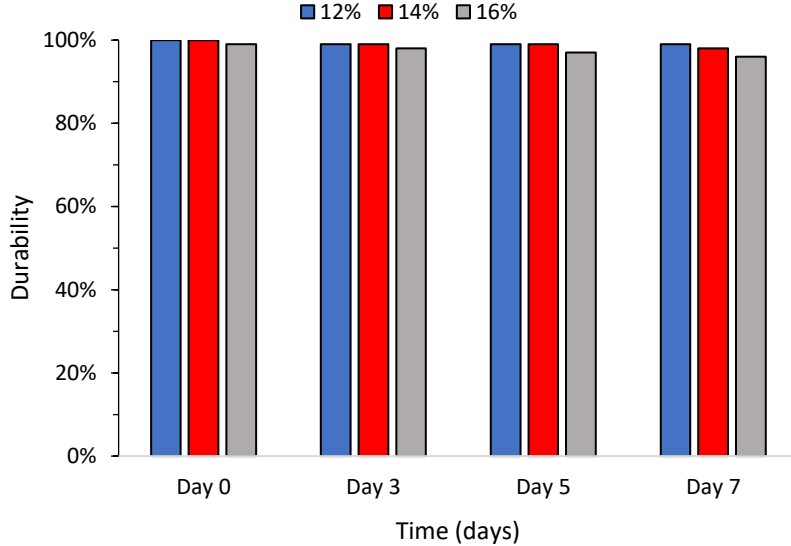

**Figure 2.** Durability study of rice husk briquette at various moisture contents.

### 3.2.3. Compressive Strength

The data on compressive strength summarized in Table 3 indicate that the produced briquette samples survived any deformation, either in the form of fracture or breaking, under the maximum load of 5 kn, as applied by the universal testing machine. This superior compressive strength can be attributed to the high applied pressure of 150 MPa during the compression stage. The maximum load applied in the study was also high, suggesting the high quality of the produced briquettes. The results suggest that the produced briquette can be labelled as high durability with good compressive strength, which was also reported from another study for briquette formation using rice husk [35]. The data of the compression test in Table 3 do not clearly show the relative strength of the briquettes prepared from different moisture contents. All sample could withstand the load of 500 kgf. However, one can deduce the relative strength of the samples from other properties, such as the pore volume and porosity, as detailed in another report [36].

**Table 3.** Compressive strength study for rice husk briquette at various moisture contents.

|  | Length (mm) | Thickness (mm) | Load @ 8 mm (kgf) |
| --- | --- | --- | --- |
| 12% | 40 | 9 | 500 |
| 14% | 40 | 9 | 500 |
| 16% | 40 | 9 | 500 |

### 3.2.4. Surface Morphology

The surface morphologies of the briquette samples prepared from a blend under various moisture contents are shown in Figure 3. SEM images are useful to observe the binding of particles, and voids/pores formed in the briquette structure. Figure 3 shows that the bonding between the particles was mainly solid bridges. It was reported earlier that the bonding force between the particles was affected by the process variables of a standard biomass solid, such as moisture content, binder, temperature, and pressure [37]. According to the microstructure images in Figure 3, there was no obvious pores on the surface. However, as shown from the highest heating value (HHV) measurements, the briquette prepared from the blend at a moisture content of 14% is the most attractive. We believe that it possesses an optimum pore size, which allows the oxygen from the air to diffuse through the pore and sustain better combustion, with longer ignition compared to the rest.

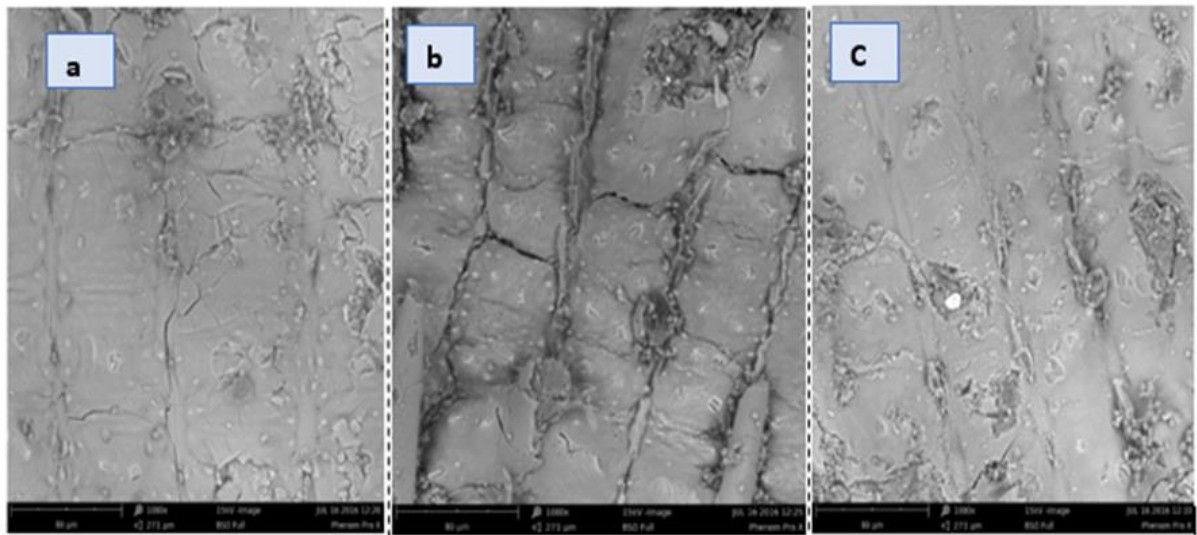

**Figure 3.** Surface morphology of rice husk briquette: (**a**) 12%, (**b**) 14%, and (**c**) 16% moisture content.

### 3.3. Chemical Properties

3.3.1. Proximate and Ultimate Analysis

The rice husk briquette at various moisture contents showed different elemental compositions, as the moisture content influenced the composition, as shown in Table 4. Rice husk blend showed a lower elemental percentage compared to rice husk briquette, mainly in carbon content, which was the most crucial element, as it affected combustion behavior. According to biomass residues it contained a high amount of carbon and hydrogen. The elemental compositions and the HHV changed substantially when compared to the raw rice husk with the blend rice husk. Those changes can be attributed to the different compositions and the HHV of the kraft lignin used as the binder, which accounted for 10% wt. Analysis of the composition and HHV of the blend and the briquette shows that they did not differ significantly, and suggesting that no reaction occurred during mixing, storage, and briquetting that could affect the material composition and hence the HHV.

**Table 4.** The ultimate and proximate analysis of rice husk blend and briquette at a various moisture contents.

| MC | Ultimate Analysis | | | | | Proximate Analysis | | | |
|---|---|---|---|---|---|---|---|---|---|
| | C (%) | H (%) | N (%) | S (%) | O (%) | VM (%) | FC (%) | ASH (%) | HHV (MJ/kg) |
| 10% Raw RH | 35.82 | 6.15 | 5.57 | 0.52 | 51.95 | nd | nd | nd | 8.97 |
| **Rice Husk Blend** | | | | | | | | | |
| 12% | 36.63 | 5.89 | 5.96 | 0.82 | 50.70 | 40 | 45.53 | 4.92 | 13.87 |
| 14% | 34.46 | 5.55 | 5.63 | 1.00 | 53.33 | 30 | 55.93 | 4.70 | 14.23 |
| 16% | 34.42 | 5.54 | 5.93 | 1.17 | 52.94 | 39 | 47.73 | 5.27 | 13.08 |
| **Rice Husk Briquette** | | | | | | | | | |
| 12% | 39.43 | 5.87 | 6.36 | 0.81 | 48.43 | 42.00 | 45.79 | 4.21 | 14.04 |
| 14% | 41.23 | 5.71 | 5.81 | 0.73 | 46.55 | 40.50 | 46.00 | 3.97 | 17.69 |
| 16% | 38.76 | 5.99 | 5.54 | 0.78 | 49.00 | 41.00 | 46.37 | 4.63 | 13.10 |

MC = moisture content, C = carbon, H = hydrogen, N = nitrogen, S = sulphur, O = oxygen, VM = volatile matter, FC = fixed carbon, HHV = higher heating value, RH = rice husk, nd = no detectable.

Generally, high carbon content influences the combustion behavior, which affects the ash fusion [38]. Under the proximate analysis study, ash content was the major concern, as

it is one of the limitations of using rice husk as a solid fuel. Ash content is predominant inorganic substance obtained after complete combustion of biomass takes place [39]. Typically, ash content contains calcium, potassium, magnesium, and phosphorus. One problem that hinders the thermal process of biomass is the deposition and agglomeration caused by minerals in ash melts [40]. Therefore, a higher fixed carbon content promotes higher calorific energy in the charcoal produced. Additionally, a higher fixed carbon content results in a lower volatile matter content. The overall mass of the briquette decreases as the volatile combustion phase takes place. Increasing hydrogen to carbon ratio augmented the combustion, although to a lesser extent, as the carbon ratio rises. However, biomass with high volatile content is able to lose 90% of its initial mass at the devotilization stage of combustion. Hence, from the proximate and ultimate analysis obtained it was crucial to make a comparison of other studies that have been done with this project, mainly of the calorific value and thermogravimetric analysis [41].

### 3.3.2. Calorific Value

Figure 4 shows the higher heating value of rice husk blend and briquette at various moisture contents. At moisture content 14%, the highest heating values were 14.23 MJ/kg before briquetting, and 17.69 MJ/kg after briquetting. While at the moisture contents of 12% and 16% the heating value raised significantly from 13.87 MJ/kg to 14.04 MJ/kg and 13.08 MJ/kg to 13.11 MJ/kg, respectively. Since the rice husk briquette at a moisture content of 14% produced the highest heating value, it was selected as the best rice husk briquette developed in this study.

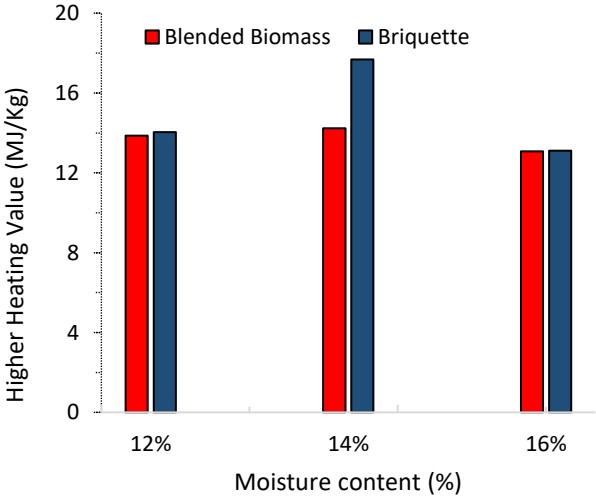

**Figure 4.** The higher heating value of rice husk blend and briquette at a various moisture contents.

The heating value increases as the wet basis of the briquette reduces, while the moisture content must be at an optimum level for the briquette to have good combustive behavior [42]. High carbon and hydrogen component presence promotes a better heating value, while inversely a high ash content gives a lower heating value. Besides that, the binding agent may typically influence the incremental heating value [43]. The best durability also gives a significantly higher heating value when combined with optimum pore volume, which allows the gases to pass through the pore to supply oxygen for the combustion reaction. Consequently, all the briquettes produced at various moisture contents were able to promote their respective heating values according to their properties.

### 3.3.3. Thermogravimetric Analysis

Figures 5–7 show the thermogravimetric analysis graph plotted for sample temperature concerning weight percentage and derivative weight percentage for all rice husk blends and briquettes at various moisture contents. Thermogravimetric analysis was per-

formed for rice husk briquette to determine the degradation profile, without limitation in heat and mass transfer of a low heating rate. From the sample temperature versus weight percentage weight plots, the briquette volatile matter, fixed carbon, and moisture can be determined. It can also be assigned as the degradation temperature of rice husk. Besides this, from the sample temperature versus the derivative weight percentage plot, the degradation temperature of the briquette moisture content, volatile matter, and fixed carbon can be determined. However, the focus of this study was to determine the rice husk blend and briquette degradation temperature. Thus, for a clear observation of the degradation temperature, the value was extracted from the graph summarized in Table 5.

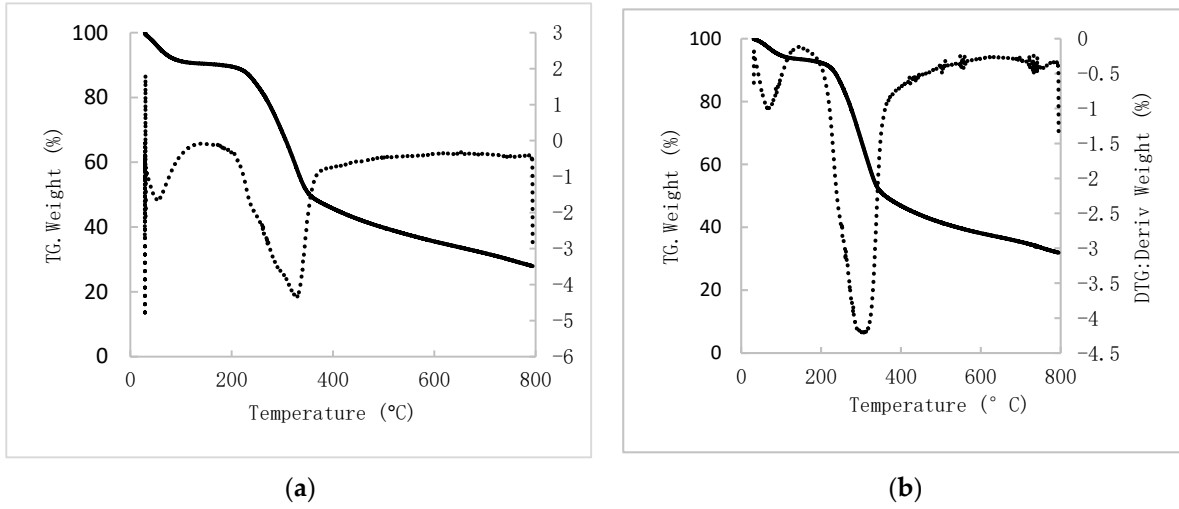

(**a**)                (**b**)

**Figure 5.** Thermogravimetric analysis (TGA) and derivative thermogravimetric analysis (DTA) analysis of (**a**) rice husk blend, and (**b**) rice husk briquette at 12% moisture content.

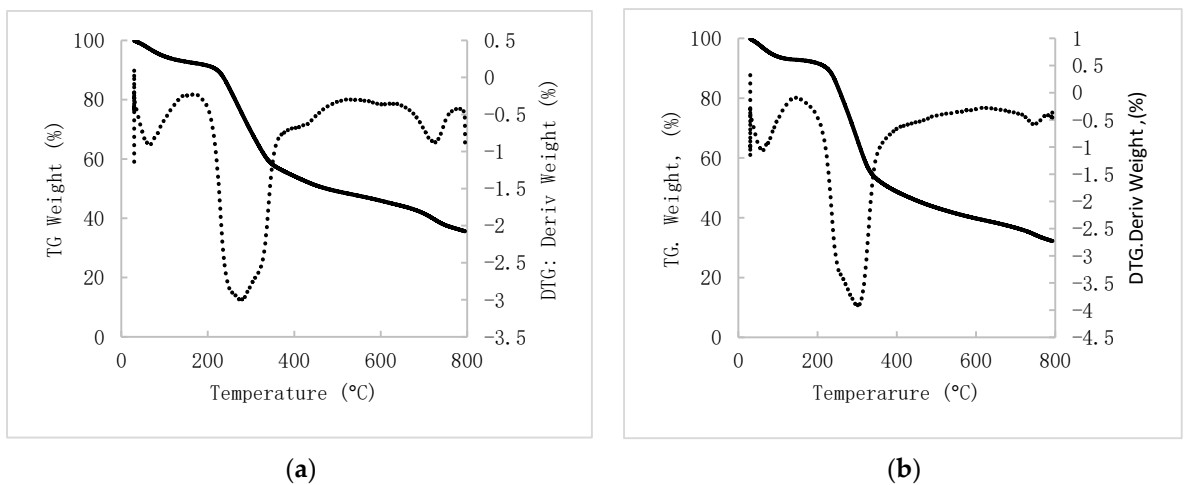

(**a**)                (**b**)

**Figure 6.** TGA and DTA analysis of (**a**) rice husk blend, and (**b**) rice husk briquette at 14% moisture content.

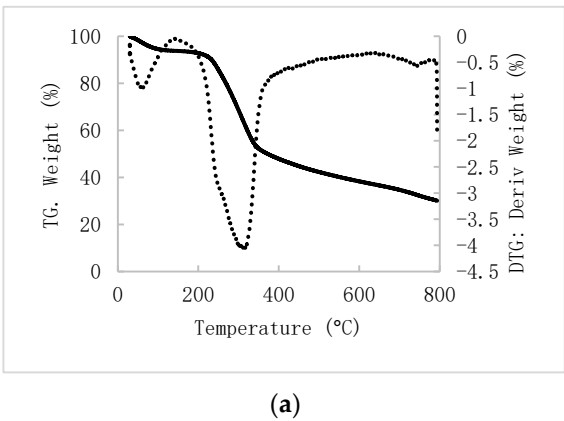 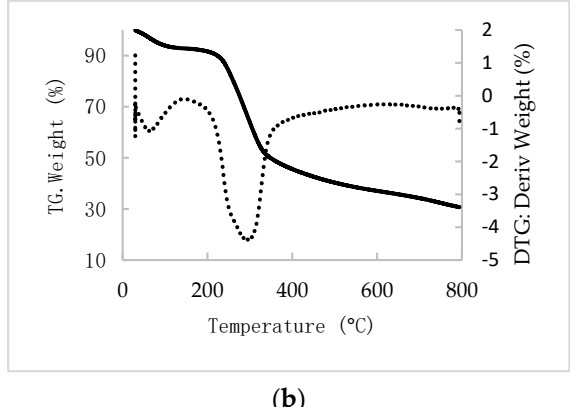

|  (a)  |  (b)  |
|:---:|:---:|

**Figure 7.** TGA and DTA analysis of (**a**) rice husk blend, and (**b**) rice husk briquette at 16% moisture content.

**Table 5.** The thermogravimetric analysis of blended rice husk and briquette at various moisture contents.

| | Degradation Temperature°C | | |
|---|:---:|:---:|:---:|
| Moisture Content | 12% | 14% | 16% |
| Blend | 330 ± 10 | 270 ± 10 | 320 ± 10 |
| Briquette | 300 ± 10 | 300 ± 10 | 300 ± 10 |

According to the results shown in Table 5, a small weight loss for the rice husk blends occurred at a temperature range of 50–200 °C, and maximum weight loss occurred at a temperature range of 250–330 °C for all samples. A rather higher range of small weight loss was observed for the rice husk briquette at 50–220 °C, and the peak weight loss occurred at a temperature of 300 °C.

Before analysis, the degradation temperature varied before rice husks were compacted into briquettes, where at moisture content 14% the degradation occurred earliest at a temperature 270 ± 10 °C, compared to the rice husk blend at moisture contents 12% and 16%, which degraded at temperatures 330 ± 10 °C and 320 ± 10 °C, respectively. This result indicates that lower temperatures are required for the water content of the rice husk blend at a moisture content of 14% to start vaporizing, which promotes a better heating value than a rice husk blend at moisture contents of 12% and 16%, as shown in Figure 4. However, the degradation temperature for the rice husk briquettes at all moisture contents was at 300 ± 10 °C. This was due to the durability of the briquettes obtained being similar, and above the standard pellet and briquette quality [44]. The range of moisture content difference was only 4% between all the briquettes, allowing the degradation temperature to be similar. Therefore, the rice husk briquettes of all moisture contents are degraded at the same temperature, but the heating value differs.

### 3.4. Comparison with Agricultural Biomass Residue-Based Briquette

Table 6 summarizes the typical moisture content of a few agricultural residues which have been widely used in energy applications. Moisture content in a material is affected by many factors. Generally, biomass has two types of moisture content (intrinsic and extrinsic) [45]. Intrinsic moisture content, also called bound water, is the moisture from the biomass itself. It is not affected by external factors. On the other hand, extrinsic moisture is the moisture which can be changed, depending on weather conditions during harvesting. The moisture content of agricultural waste residues such as husk and straw falls from below 20% to up to 60% for bagasse, while the moisture content of forest residue falls between the range 40 and 50% [46].

As shown in Table 6, the moisture content of biomass residue varies accordingly in different parts and catalogs. According to L. Chena et al. [47], the moisture content is best to be low, in a range of 10–15%, as high moisture content would require energy

for drying prior to being used as a fuel, which was also reported in [48]. Besides that, optimum moisture content for cellulosic biomass falls within the range of 8–12% [49]. Feedstock moisture content is essential, as it is the most important variable for controlling the moisture uptake in the production of pellets [50]. The comparison results show that the rice husk blend with kraft lignin is highly durable at all moisture contents, where the kraft lignin works as a binder, which makes a huge impact on the production of solid fuel. The calorific values of the samples reported in Table 6 were probably determined by the same technology, that is a bomb calorimeter. It was found that the highest calorific values, shown as highest heating value (HHV) in Table 6, were rice bran (20.5 MJ/kg), followed by coffee husk (19.35 MJ/kg), oat straw (17.54 MJ/kg), corn cob (17.25 MJ/kg), rice straw (17.24 MJ/kg), groundnut straw (17.10 MJ/kg), coconut shell (17.9 MJ/kg), and cotton stalk (16.23 MJ/kg) compared to other agricultural biomass residue which had calorific values less than 15 MJ/kg. From Table 6, it can be noticed that the high ash content of agricultural waste affects the heating value of briquette, which makes it less desirable as fuel, whereas lower ash content adds to its desirability [51].

Ash content is the remainder of the mass of inorganic material that remains after complete burning under specific conditions, as a fraction of the original mass [52]. Biomass combustion can form ash, and the major ash-forming elements in biomass are Na, Ca, K, Mg, and various heavy metals. Therefore, the chemical composition of biomass materials has a significant effect on the ash content [24]. The highest ash content is for rice husk at 19.5% obtained from this study, followed by rice straw (11.25%), wheat straw (12.8%), corn stalk (10%), and other agricultural biomass that has an ash content of below 9%, as demonstrated in Table 6.

**Table 6.** Comparison of agricultural biomass residue with rice husk–kraft lignin briquettes.

| Crop | Residue | Moisture Content (%) | Ash content (%) | Volatile Matter (%) | Fixed Carbon | HHV (MJ/kg) | Relaxed Density (kg/m$^3$) | Ref. |
|---|---|---|---|---|---|---|---|---|
| Rice | Straw | 12.2 | 11.25 | 42.2 | 50.3 | 16.1 | 1011.23 | [53] |
| Rice | bran | 12.5 | 8.7 | 44.51 | 55.01 | 20.5 | 887.25 | [53] |
| Corn | Stalk | 16.8 | 10 | 61.20 | 36.4 | 13.79 | 1013.22 | [54] |
| Corn | Cob | 7.7 | 8.1 | 64.2 | 33.21 | 17.25 | 1007.32 | [55] |
| Corn | Husk | 11 | 2.84 | 62.98 | 36.15 | 14.44 | 1035.30 | [56] |
| Wheat | Straw | 14.6 | 12.8 | 55.21 | 37.21 | 17.9 | 1022.40 | [57] |
| Oats | Straw | 15 | na | 45.2- | 39.51 | 17.54 | 955.24 | [58] |
| Barley | Straw | 16.3 | 4.3 | 44.52 | 41.21 | 14.85 | 1011.21 | [58] |
| Sorghum | Straw | 15 | na | 46.35 | 52.35 | 14.25 | 956.34 | [31] |
| Cassava | Stalk | 15 | na | 42.30 | 42.1 | 12.79 | 927.35 | [59] |
| Groundnut | Husk/Shell | 7.8 | na | 57.1 | 46.43 | 13.15 | 957.25 | [60] |
| Groundnut | Straw | 12 | 1.3 | 43.25 | 39.45 | 17.10 | 834.49 | [60] |
| Soybean | Straw | 15 | na | 48.21 | 47.34 | 14.21 | 954.50 | [61] |
| Sugar cane | Bagasse | 49.8 | 6 | 52.40 | 41.12 | 11.25 | 827.35 | [31] |
| Sugar cane | Tops/Leaves | 62.5 | 1.2 | 54.35 | 38.5 | 13.45 | 987.25 | [62] |
| Cotton | Stalk | 12 | 5.4 | 55.21 | 51.3 | 14.23 | 1157 | [63] |
| Cotton | Husk | 10 | 6 | 49.34 | | 13.35 | 1153 | [64] |
| Coconut | Shell | 10.9 | 1 | 49.25 | 54.3 | 17.24 | 875.43 | [62] |
| Oil palm | Shell | 12.3 | 4 | 46.23 | 44.35 | 14.45 | 955.40 | [65] |
| Oil palm | Fibre | 36.7 | 5 | 41.34 | 43.2 | | 855.59 | [65] |
| Oil palm | Empty bunches | 36.7 | 5 | 43.35 | 39.5 | 16.30 | 947.15 | [65] |
| Coffee | Husk | 15 | na | 57.25 | 42.10 | 19.35 | 1132 | [66] |
| Rice | Husk | 10 | 19.5 | nd | 35.82 | 8.97 | 543 | This study |
| Rice | Husk blen d+ kraft lignin | 12 | 4.92 | 40 | 36.63 | 13.868 | 1136 | This study |
| | | 14 | 4.70 | 30 | 34.46 | 14.229 | 825 | |
| | | 16 | 5.27 | 39 | 34.42 | 13.08 | 673 | |
| Rice | Husk Briquette | 12 | 4.21 | 42 | 39.43 | 14.04 | 1021 | This study |
| | | 14 | 3.97 | 40.5 | 41.23 | 17.688 | 801 | |
| | | 16 | 4.63 | 41 | 38.76 | 13.106 | 644 | |

Our overall findings suggest the importance of water content in determining the properties of the briquette product. Due to variation of the water contents of the rice husk materials, the findings imply the need for water content characterization and modification of the rice husk before the briquetting process. Thanks to the desirable properties of the rice husk–kraft lignin briquettes, the economic value of the agricultural biomass waste can be enhanced and converted into a valuable product. This lightens the burden on the environment by reducing waste, and reducing the consumption of fossil-based fuels, which can be substituted by the briquettes.

## 4. Conclusions

Rice husk briquettes blended with kraft lignin were efficiently formed at various moisture contents of 12, 14, and 16%. They displayed highly durable properties, accompanied by a good compressive strength after being forced under 5 kN without any cracking or breaking occurring. The briquette with a moisture content of 14% possessed the standard density requirements for briquettes, at $825 \pm 0.26$ kg/m$^3$, along with an optimum pore size. It also had the highest heating value of 17.688 MJ/Kg.

**Author Contributions:** Conceptualization, A.A.H.S. and N.Y.H.; methodology, N.Y.H., V.D.V. and A.A.H.S.; software, A.A.H.S. and F.A.S.R.; validation, A.A.H.S., S.A.R. and N.Y.H.; formal analysis, A.A.H.S.; investigation, A.A.H.S. and N.Y.H.; resources, N.Y.H.; data curation, A.A.H.S., H.K.A. and N.Y.H.; writing—original draft preparation, A.A.H.S. and V.D.V.; writing—review and editing, A.A.H.S., V.D.V., M.R.B. and F.A.S.R.; visualization, A.A.H.S. and N.Y.H.; supervision, N.Y.H., A.M.P. and M.T.A.; project administration, N.Y.H. and A.A.H.S.; funding acquisition, N.Y.H. All authors have read and agreed to the published version of the manuscript.

**Funding:** This work was funded by the Yayasan Universiti Teknologi PETRONAS (YUTP) with grant code YUTP-015LC0-210.

**Institutional Review Board Statement:** Not applicable.

**Informed Consent Statement:** Not applicable.

**Data Availability Statement:** The data presented in this study are available from the corresponding author upon a reasonable request.

**Acknowledgments:** The authors would like to acknowledge the efforts of the Universiti Teknologi Petronas Malaysia for financing the project through the Yayasan Universiti Teknologi PETRONAS (YUTP) with grant code YUTP-015LC0-210.

**Conflicts of Interest:** The authors declare no conflict of interest.

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
