# Peer review of "Moisture Content Impact on Properties of Briquette Produced from Rice Husk Waste"

_sustainability, doi:10.3390/su13063069_

Round 1

Reviewer 1 Report

I have some questions:

Questions:

  1. Chap. 2.3.4. Surface morphology....add the conditions of measurement.
  2. What kind of kraft lignin binder you used to ease the compaction of rice husk briquette?
  1. Is the company interested in this kind of briquettes from rise husk?
  1. It would be good to add the part on the impact production of briquettes on the environment.

Author Response

Response to Editor and Reviewers

The authors would like to thank the reviewers for the valuable comments and suggestions that contribute to improving our manuscript. Amendments have been provided accordingly, and all modifications are highlighted in red in the revised version of the manuscript.

Detailed point-by-point responses to the reviewer’s comments are given below.

Reviewer 1

Comments and Suggestions for Authors:  I have some questions.

Response to Reviewer: Thank you for your valuable review.

Questions and comments

Comment 1: In chap. 2.3.4. Surface morphology.... add the conditions of measurement.

Response 1: Conditions of measurement of surface morphology have been added including the scale and shown in Figure 3. Information about SEM sample pre-treatment has also been provided in Section 2.3.4 in the revised manuscript.

Comment 2: What kind of kraft lignin binder you used to ease the compaction of rice husk briquette?

Response 2: The kraft lignin binder was prepared according to Reference 20. The information has now been included in Section 2.3.3 of the revised manuscript.  

Comments 3:  Is the company interested in this kind of briquettes from rise husk?

Response 3: Yes, the company is interested in briquettes produced from rice husk due to the abundance of rice husk in Malaysia, and this merit will let kind of briquettes had an economical value added to the products.

Comments 4: It would be good to add the part on the impact production of briquettes on the environment.

Response 4: Thank you for the suggestion. The environmental impact of briquettes on the environment has now been included in the manuscript (see the last paragraph of Section 3.4).

Reviewer 2 Report

The paper investigates the effect of initial moisture contents of rice husk-based briquette blended with kraft lignin on chemical and physical characteristics. This study proved the approach to enhance the quality of briquette from rice busk by controlling of moisture content. The topic of the paper is interesting and is fully in line with the aims and scope of the Journal.

However, it contains some lacks and elements that need to be completed. Below the main lacks and dubious elements that need to explain and/or extended:

  1. Table 1, the analysis is dry or wet basis? Clarify and mark it in the table
  2. Equation 2 is not clearly presented, is it y-x/m?
  3. Line 140, Does it mean this study also take the oven for drying in the oven? In addition, the moisture content in the reference is different compared to this study.
  4. Line 163-164 the smaller particle size of biomass residue promotes higher density under high pressures as reported earlier. Missing reference.
  5. Line 167-169, The sentences should be reorganized here to be logical. Mention the standard first and then select the two samples. The reference of standard should be given.

“The standard briquette density falls within the range of 750 kg/m3  and  above. Based on the findings, briquette samples prepared from 167 blends with moisture contents of 12 % and 14% are identified suitable as they meet the standard briquette density requirement.”

  1. Line 171, Keep the number of decimals for relaxed density the same.
  2. Line 178, wrong spelling. tended s should be tends
  3. Line 184, reference for the durability range 95-100%
  4. Line 200, Table 3 is not well explained in the text. What result can we get from Table 3 about compressive strength study? The last column should be explained in the text.
  5. Line 207 temperatures and pressure, no comma
  6. Line 208-210, the author should give the pore size range instead of only saying smaller, medium, uneven size.
  7. Line 214, I think the scale of SEM figure should be given to know the pore size.
  8. Line 210. The reason of why medium size is attractive than smaller and uneven is not clearly explained in my opinion, If it is the previous conclusion, then give the reference.
  9. Line 222, wrong spelling, with the blend rice husk,
  10. Line 241, Table 4, keep the decimals the same in the same column
  11. Line 251, I think you can only get the conclusion of best option of 14% in terms of HHV in this sub-section. The best briquette in the whole study should combine the conclusion from other sections.
  12. Improvement of Figure 5-7:
  • Tick marks are used for the right vertical scale but not for left. Try to add for left vertical scale and horizontal scale.
  • There is a letter “h” on the left in Figure 6-b.
  • Suggestion to combine Figure 5a, 6a and 7a into one figure for better comparison between moisture content. Combine Figure 5b,6b and 7b into one figure.
  1. Line 291, wrong spelling, less temperatures. It should be “lower temperature is required”
  2. Line 301 , widely used
  3. Line 303-305, wrong spelling and unclear sentence’s structure. Suggestion” Intrinsic moisture content, also called bound water, is the moisture from biomass itself. It does not affect by external factors. On the other hand, extrinsic moisture is the moist which can be changed, prevailing to weather conditions during harvesting…..”
  4. Line 308, wrong spelling. “Biomass residue varies”
  5. Line 310-312. The sentence before and after “Additionally” are actually the same reason for why high moisture is not favored. Because you need extra energy for drying the water. Unnecessary to make it as two reasons.

Author Response

Response to Editor and Reviewers

The authors would like to thank the reviewers for the valuable comments and suggestions that contribute to improve our manuscript. Amendments were provided accordingly, and all modifications were highlighted red track in the revised version of the manuscript. Response to each comment, point by point, is given below.

Reviewer 2

 Comments and Suggestions for Authors: The paper investigates the effect of initial moisture contents of rice husk-based briquette blended with kraft lignin on chemical and physical characteristics. This study proved the approach to enhance the quality of briquette from rice busk by controlling of moisture content. The topic of the paper is interesting and is fully in line with the aims and scope of the Journal. However, it contains some lacks and elements that need to be completed. Below the main lacks and dubious elements that need to explain and/or extended:

Response: Thank you for your valuable comments and suggestions, we have revised the manuscript to accommodate the comments and suggestion by the reviewer.

Following are the comments:

No

Section

Comments

Response to reviewer comments

1

 Introduction

Table 1, the analysis is dry or wet basis? Clarify and mark it in the table

The analysis is dry basis and has been

amended. See Table 1 in the revised manuscript.

2

Materials and Methods

 Equation 2 is not clearly presented, is it y-x/m?

Equations 2 has been rewritten to improve the clarity. See page 4.

Results and Discussion

Line 140, Does it mean this study also take the oven for drying in the oven? In addition, the moisture content in the reference is different compared to this study

We mean Briquette Compression process. The method adopted is similar to the one by Ndindeng et al.

The sentences have been revised accordingly. Please see the revised manuscript (Section 3.2, first paragraph).

4

 Results and Discussion

Line 163-164 the smaller particle size of biomass residue promotes higher density under high pressures as reported earlier. Missing reference

The reference has been added as recommended, please see the revised manuscript (line 172).

5

 Results and Discussion

Line 167-169, The sentences should be reorganized here to be logical. Mention the standard first
and then select the two samples. The reference of standard should be given. “The standard briquette density falls within the range of 750 kg/m3 and above. Based on the
findings, briquette samples prepared from 167 blends with moisture contents of 12 % and 14%
are identified suitable as they meet the standard briquette density requirement.”

The sentences have been reorganized accordingly. Please see the revised manuscript (Section 3.2.1, second paragraph).

6

 Results and Discussion

Line 171, Keep the number of decimals for relaxed density the same

Complied as recommended. See Table 2 in the revised manuscript

7

Results and Discussion

Line 178, wrong spelling. tended s should be tends

Complied as recommended.

8

Results and Discussion

Line 184, reference for the durability range 95-100%

The reference has been added as recommended, please see the revised manuscript (Ref #35).

9

Results and Discussion

Line 200, Table 3 is not well explained in the text. What result can we get from Table 3 about
compressive strength study? The last column should be explained in the text

Discussion on data in Table 3 has been extended. See Section 3.2.3 in the revised manuscript.

10

Results and Discussion

Line 207 temperatures and pressure, no comma

Complied as recommended.

11

Results and Discussion

Line 208-210, the author should give the pore size range instead of only saying smaller, medium,
uneven size

The ranges of pore size mentioned in the manuscript was based on interpretation of Figure 3, not from measurement. So we can not provide the actual value. To avoid confusion, we omit the part in the revised manuscript (See section 3.2.4).

12

Results and Discussion

Line 214, I think the scale of SEM figure should be given to know the pore size

surface morphology Figure 3 had been redrawn and shown the measurement scales.

13

Results and Discussion

Line 210. The reason of why medium size is attractive than smaller and uneven is not clearly explained in my opinion, if it is the previous conclusion, then give the reference

Please refer to our response to comment 11.

14

Results and Discussion

Line 222, wrong spelling, with the blend rice husk

Complied as recommended.

15

Results and Discussion

Line 241, Table 4, keep the decimals the same in the same column

Complied as recommended.

16

Results and Discussion

Line 251, I think you can only get the conclusion of best option of 14% in terms of HHV in this sub-section. The best briquette in the whole study should combine the conclusion from other sections.

Thank you for the suggestion, we agree and revised the manuscript accordingly.

17

Results and Discussion

Improvement of Figure 5-7:
1) Tick marks are used for the right vertical scale but not for left. Try to add for left vertical
scale and horizontal scale.
2) There is a letter “h” on the left in Figure 6-b.
3) Suggestion to combine Figure 5a, 6a and 7a into one figure for better comparison between
moisture content. Combine Figure 5b,6b and 7b into one figure

All figures have been improved for clarity. Thank you for the suggestion on the figures layout. We opt to maintain the figure Table 5.

18

Results and Discussion

Line 291, wrong spelling, less temperatures. It should be “lower temperature is required”

It has been amended accordingly.

19

Results and Discussion

Line 301, widely used

It has been amended accordingly.

20

Results and Discussion

Line 303-305, wrong spelling, and unclear sentence’s structure. Suggestion” Intrinsic moisture
content, also called bound water, is the moisture from biomass itself. It does not affect by external factors. On the other hand, extrinsic moisture is the moist which can be changed, prevailing to weather conditions during harvesting…..”

It has been amended accordingly. See Section 3.4 first paragraph.

21

Results and Discussion

Line 308, wrong spelling. “Biomass residue varies”

It has been amended accordingly.

22

Results and Discussion

Line 310-312. The sentence before and after “Additionally” are actually the same reason for why high moisture is not favored. Because you need extra energy for drying the water. Unnecessary to make it as two reasons.

It has been amended accordingly. Redundancy has been omitted. See Section 3.4, second paragraph.

Reviewer 3 Report

3: I suggest changing wastes to waste in the title.

23: Convert MJ / Kg to MJ⸳kg-1. I suggest to use the notation throughout the work.

66: Please add your work objective at the end of the Introduction chapter.

122: 2g - please make a space.

137-138: The form of all briquette samples is good 138 thanks to a good combination of materials, blend ratio, applied pressure, and the biomass size. - Style to improve.

146: Table 2 summarizes the properties of…. - Style to improve.

274: Figure 5,6,7 – Please standardize the notation, font, etc.

320: (17.25MJ/kg) – please make a space.

Author Response

 Response to Editor and Reviewers

The authors would like to thank the reviewers for the valuable comments and suggestions that contribute to improving our manuscript. Amendments were provided accordingly, and all modifications were highlighted in red track in the revised version of the manuscript. Response to the comments is given below.

Reviewer 3

Comments and Suggestions

Comment 1: I suggest changing wastes to waste in the title.

Response 1: Thank you. The manuscript title has been revised as you suggested.

Comment 2: Convert MJ / Kg to MJ⸳kg-1. I suggest using notation throughout the work (line 23).

Response 2: Amended as recommended. See the revised manuscript (line 25).

Comment 3: Please add your work objective at the end of the Introduction chapter (line 66)

Response 3: Amended as recommended. See the revised manuscript (line 65-68).

Comment 4:2g please make a space (line 122).

Response 4: Amended as suggested.

Comment 5: The form of all briquette samples is good 138 thanks to a good combination of materials, blend ratio, applied pressure, and biomass size. - Style to improve (line 137-138).

Response 5: Complied as recommended. See the revised manuscript (line 142-143).

Comment 6: Table 2summarizes the properties of…. - Style to improve ((line 146).

Response 6: Complied as recommended. See the revised manuscript (line 152-153).

Comment 7: In line 274: Figure 5,6,7 – Please standardize the notation, font, etc.

Response 7: Complied as suggested, please check the updated figures in the the revised manuscript.

Comment 8: In line 320: (17.25MJ/kg) – please make a space.

Response 8: Complied as suggested, please see line 327.
